# Kawasaki Disease and Vaccination: Prospective Case-Control and Case-Crossover Studies among Infants in Japan

**DOI:** 10.3390/vaccines9080839

**Published:** 2021-07-30

**Authors:** Kenji Murata, Sagano Onoyama, Kenichiro Yamamura, Yumi Mizuno, Kenji Furuno, Keita Matsubara, Ken Hatae, Kiminori Masuda, Yuichi Nomura, Takuro Ohno, Akiko Kinumaki, Masaru Miura, Yasunari Sakai, Shouichi Ohga, Wakaba Fukushima, Junji Kishimoto, Yosikazu Nakamura, Toshiro Hara

**Affiliations:** 1Kawasaki Disease Center, Fukuoka Children’s Hospital, Fukuoka 813–0017, Japan; murata.k@fcho.jp (K.M.); sonoyama178@gmail.com (S.O.); mizuno.y@fcho.jp (Y.M.); furuno.k@fcho.jp (K.F.); 2Department of Pediatrics, Japanese Red Cross Fukuoka Hospital, Fukuoka 815–8555, Japan; hatae.k@fukuoka-med.jrc.or.jp; 3Department of Perinatal and Pediatric Medicine, Graduate School of Medical Sciences, Kyushu University, Fukuoka 812–8582, Japan; yamamura@pediatr.med.kyushu-u.ac.jp; 4Department of Pediatrics, Hiroshima City Funairi Citizens Hospital, Hiroshima 730–0844, Japan; keitamatsubara3@gmail.com; 5Department of Pediatrics, Kagoshima City Hospital, Kagoshima 890–8760, Japan; kiminori1123@yahoo.co.jp (K.M.); uichiyuichi@gmail.com (Y.N.); 6Department of Pediatrics, Oita Prefectural Hospital, Oita 870–8511, Japan; ohnot55@yahoo.co.jp; 7Department of General Pediatrics, Tokyo Metropolitan Children’s Medical Center, Fuchu, Tokyo 183–8561, Japan; kinumaki-tky@umin.ac.jp; 8Department of Cardiology, Tokyo Metropolitan Children’s Medical Center, Fuchu, Tokyo 183–8561, Japan; masaru10miura@gmail.com; 9Department of Pediatrics, Graduate School of Medical Sciences, Kyushu University, Fukuoka 812–8582, Japan; yasunari@med.kyushu-u.ac.jp (Y.S.); ohgas@pediatr.med.kyushu-u.ac.jp (S.O.); 10Department of Public Health, Research Center for Infectious Disease Sciences, Osaka City University Graduate School of Medicine, Osaka City University, Osaka 558–8585, Japan; wakaba@med.osaka-cu.ac.jp; 11Center for Clinical and Translational Research, Kyushu University Hospital, Fukuoka 812–8582, Japan; j_kishi@digital.med.kyushu-u.ac.jp; 12Department of Public Health, Jichi Medical University, Shimotsuke 329–0498, Japan; nakamuyk@jichi.ac.jp

**Keywords:** Kawasaki disease, vaccines, infant, case-control study, case-crossover study, Nationwide Survey of Kawasaki disease in Japan

## Abstract

The causal effects of vaccines on Kawasaki disease (KD) remain elusive. We aimed to examine the association between vaccines administered during infancy and the development of KD in Japan. We conducted a multicenter prospective case-control study using questionnaires and compared the vaccination status of infants (age: 6 weeks to 9 months) who developed KD (KD group; *n* = 102) and those who did not develop KD (non-KD group; *n* = 139). Next, we performed a case-crossover study of 98 cases in the KD group and compared the status of vaccinations between the case and control periods. We also compared the incidence of KD in children for each 5-year period before and after the addition of new vaccines (2012–2013) using data from the Nationwide Survey of KD. In the case-control study, the vaccination status of the KD and control groups did not differ to a statistically significant extent. Multivariable analysis of the vaccination status and patient backgrounds showed no significant association between vaccination and KD development. In the case-crossover study, the status of vaccinations during the case and control periods did not differ to a statistically significant extent. In the analysis of data from the Nationwide Survey of KD, the incidence of KD in children of ages subject to frequent vaccination showed no significant increases in the latter five years, 2014–2018. Based on these prospective analyses, we confirmed that vaccination in early infancy did not affect the risk of KD.

## 1. Introduction

Kawasaki disease (KD) is an acute systemic vasculitis of unknown cause in childhood [1]. Coronary artery abnormalities develop in 15–25% of untreated children with KD, and the sequelae include giant coronary aneurysms and coronary artery stenosis, leading to myocardial infarction and death [2]. The clinical and epidemiological features support that the disease is induced by an infectious agent in genetically susceptible individuals [3,4,5,6,7].

Although numerous microbes have been proposed as etiologic agents of KD, few have been consistently associated with the illness [4]. Similarly, there have been reports of patients developing KD after receiving rotavirus vaccine, pneumococcal conjugate vaccine (PCV), hepatitis B virus (HBV) vaccine, diphtheria, pertussis, tetanus and inactivated poliovirus vaccine (DPT-IPV), *Haemophilus influenzae* type b (Hib) vaccine and *Bacillus Calmette-Guérin* (BCG) vaccine [8]. Among them, the rotavirus vaccine RotaTeq^®^ was associated with a higher risk of KD in pre-marketing phase 3 clinical trials, and KD was then listed as one of the serious side effects of RotaTeq^®^ in 2007 [9].

However, their causal effects have not been established [8] because (1) these studies did not always clarify the effect of confounding factors, (2) they did not compare cases and controls in the same region and study period, since KD often occurs in temporal and regional clusters [10], and (3) they did not always recruit patients prospectively.

In the present study, we performed a prospective case-control study that considered confounding factors to examine the association between the vaccines administered in infancy and the development of KD. We also employed a case-crossover study and analyzed data (obtained from the Nationwide Survey of KD) from the periods before and after the addition of new vaccines in Japan.

## 2. Materials and Methods

A multicenter study was conducted at seven participating institutions in Japan: Fukuoka Children’s Hospital, Hiroshima City Funairi Citizens Hospital, Japanese Red Cross Fukuoka Hospital, Kagoshima City Hospital, Oita Prefectural Hospital, Tokyo Metropolitan Children’s Medical Center, and Fukuoka Higashi Medical Center. This study was approved by the ethics committee at each institution. Written informed consent was obtained prior to registration from the parents or guardians of the subjects.

### 2.1. Case-Control Study

#### 2.1.1. Study Design and Subjects

This multicenter prospective case-control study included infants of 6 weeks to 9 months of age who were admitted to the seven institutions with KD (case) or without KD (control) between 1 January 2017, and 30 June 2019. Since new vaccines (rotavirus, PCV13, and Hib) were added to the national vaccination schedule in 2012–2013 in Japan, and the incidence of KD continued to increase until 2018, the study population was set from 6 weeks to 9 months of age. KD was diagnosed according to the diagnostic guidelines for Kawasaki disease (5th revised edition) of Japan [11]. Cases included both complete KD (five or six of six major symptoms) and incomplete KD (four of six major symptoms with coronary artery aneurysm). Controls were selected from patients admitted to the same facility at the same time as the KD patients. The exclusion criteria for enrollment were as follows: (i) serious underlying diseases complicated by immunodeficiency, (ii) treatment with immunosuppressive drugs, such as steroids, for renal, autoimmune, hematologic, cardiac, neurological, or other diseases, (iii) a history of immunoglobulin use, and (iv) a history of hypersensitivity to vaccines or vaccine components.

The diagnoses on the admission of the control group were as follows: respiratory infection (*n* = 62; 44.6%), urinary tract infection (*n* = 44; 31.7%), skin and soft tissue infection (*n* = 7; 5.0%), intestinal infection (*n* = 3; 2.2%), bacteremia (*n* = 2; 1.4%), allergy (*n* = 5; 3.6%), infant hemangioma (*n* = 3; 2.2%), rash (*n* = 2; 1.4%), and other (*n* = 11; focus of unknown origin, viral myocarditis, West syndrome, benign infantile spasms, afebrile convulsion, intussusception, Hirschsprung’s disease, poor feeding, poor weight gain, lactose intolerance, and apparent life-threatening events).

#### 2.1.2. Questionnaires

Questionnaires were used to collect information from a subject’s physicians and parents. The questionnaire for physicians was filled out based on interviews, medical records, and the Maternal and Child Health Handbook. The questionnaire for physicians included data on age, sex, diagnosis, date of admission, date of onset, underlying disease, and vaccination history (Hib vaccine, PCV13, HBV vaccine, DPT-IPV, rotavirus vaccine, and BCG vaccine). A copy of the Maternal and Child Health Handbook was recommended for accurate confirmation of the vaccination history, which includes the status of vaccinations and the date of each vaccination. In KD cases, data were also collected on the date of the diagnosis, a number of major symptoms of KD, treatment (intravenous immunoglobulin [IVIG], aspirin, and others), IVIG response, results of blood examination on admission (percentage of neutrophils, platelet count, aspartate aminotransferase [AST], sodium, and C-reactive protein), and the echocardiographic findings of the coronary artery, including the diameters of the proximal right coronary artery (segment [seg.] (1), the left main trunk (seg. 5), and the left anterior descending artery (seg. 6). Coronary artery aneurysm was diagnosed based on the following findings: enlargement of the peripheral coronary arteries by more than 1.5 times, a maximum diameter of ≥3 mm in patients of < 5 years of age, a maximum diameter of ≥ 4 mm in patients of > 5 years of age, or Z score ≥ 2.5, and apparent lumen irregularity [2,12].

The questionnaire for the patient’s parents included information on age, sex, diagnosis, date of admission, breastfeeding, baby food, history of allergic diseases (food allergy, atopic dermatitis, and bronchial asthma), symptoms of infection (fever, cough, rhinorrhea, vomiting, and diarrhea) within 2 months before hospitalization, antibiotic use within 2 months before hospitalization, number of family members living with the patient, number of siblings, family history of KD, housing style, and occupation of the parents (based on the Japan Standard Occupational Classification [4th revision in December 1997]: (1) specialist and technical workers; (2) administrative and managerial workers; (3) clerical worker; (4) sales worker; (5) service worker; (6) security worker; (7) agriculture, forestry and fishery worker; (8) transport and communication worker; (9) production process and related worker; (10) workers not classifiable by occupation or unemployment [including homemaker]) [13].

#### 2.1.3. Statistical Analyses

Continuous variables were tested for normality using the Shapiro-Wilk test. Comparison between the two groups was performed using the Wilcoxon rank-sum test because some variables were not normally distributed. The differences in categorical variables were compared between the two groups using Fisher’s exact test. The classifications of the parents’ occupations were compared between the two groups using Pearson’s chi-squared test. A univariate logistic regression analysis was used to examine the association between each vaccine and the development of KD and to calculate crude odds ratios. A conditional logistic regression analysis was performed for each vaccination with 10 fixed variables (age, sex, breastfeeding, baby food, history of allergic disease, symptoms of infection within 2 months before hospitalization, antibiotic use within 2 months before hospitalization, siblings, family history of KD, and housing style). *p* values of < 0.05 were considered to indicate statistical significance. Statistical analyses were performed using JMP^®^ Pro version 14.0 (SAS Institute, Cary, NC, USA) and SAS (version 9.4, SAS Institute Inc., Cary, NC, USA).

### 2.2. Case-Crossover Study

We performed a case-crossover study, an analytical method used to assess the changes in risk associated with transient exposure. This is a type of self-controlled study [14,15], and each subject is under his or her own control.

The subjects of the case-crossover study were patients in the KD group of the case-control study for whom the date of the onset of KD and the date of vaccination was known. Because all subjects in the KD group were >2 months of age, the observation period was 56 days (8 weeks) before the onset of KD.

In studies on KD in siblings [16,17,18], the interval of the onset of KD in two sibling patients peaked on the same day and at less than a week to one month, suggesting the involvement of a KD trigger that mostly occurred within a 7-day incubation period and occasionally occurred within a 28-day incubation period. We, therefore, defined the “case period” as 7 days (S) or 28 days (L) before the date of the onset of KD. In contrast, the “control period” was defined as 29–35 days (S) or 29–56 days (L) before the onset of KD, which was considered to be less associated with the development of KD (Figure 1a). 

To compare the status of vaccinations in the case and control periods, we created a table that contained the number of vaccinations and person-time per subject (Figure 1b). To estimate the risk of developing KD in association with vaccination, we calculated the Mantel–Haenszel incidence ratio (IR_MH_) as the pooled estimate and its 95% confidence interval (95% CI) [14,15].

### 2.3. The Analysis of Data from the Nationwide Survey of KD in Japan

The Nationwide Survey of KD in Japan is conducted every two years, and the results are open to the public [19,20,21,22,23,24]. These include data on the number and incidence of KD patients by age group (per 100,000 children of 0–4 years of age per year in the vital statistics data for Japan) but do not include information on vaccines. In Japan, two types of rotavirus vaccines were released in November 2011 and July 2012. In 2013, the Hib vaccine and PCV13 were added to the National Immunization Program. Therefore, using Student’s *t*-test, we compared the number of KD patients and the incidence of KD in all ages and cases grouped by age (0–2 months, 3–5 months, 6–8 months, 9–11 months, and < 1 year) in the 5-year periods before and after 2012–2013.

## 3. Results

### 3.1. Case-Control Study

#### 3.1.1. Patient Characteristics 

The subjects included 241 patients (KD group, *n* = 102; control group, *n* = 139) from seven participating institutions who met the enrollment criteria. Questionnaires could be collected from the subject’s physician and parents in all cases. Eighteen patients (KD group, *n* = 8; control group, *n* = 10) had some missing data in the questionnaire items (Table 1), but vaccination history was collected without missing data from all 241 patients. The patients in the KD group (median age, 6 months; range: 2–9; standard deviation [SD]: 2.2) were significantly older than those in the control group (median age, 5 months; range, 1–9; SD, 2.3) (*p* = 0.017) (Table 1). There were 64 boys (62.7%) in the KD group and 77 boys (55.4%) in the control group; the proportions of patients of each sex did not differ between the two groups to a statistically significant extent (*p* = 0.29). The clinical features and laboratory results in the KD group are shown in Appendix A.

With regard to the patient background factors, which were determined based on the questionnaires, the proportion of patients in whom baby food feeding had been initiated in the KD group was significantly higher than that in the control group in the univariate analysis; however, the association did not remain statistically significant in the multivariable analysis (Table 1, Appendix A). The proportion of patients with an allergic disease was significantly lower in the KD group; however, there was no difference in the proportions of patients with each allergic disease (Table 1). The proportions of patients who were breast feeding, patients with a history of infection or antimicrobial therapy, patients with a family history of KD, and patients with a detached house, and the number of siblings or family members living together did not differ between the two groups to a statistically significant extent. Furthermore, there were no significant differences between the two groups in the classifications of the parents’ occupations (fathers’ occupation, *p* = 0.475; mothers’ occupation, *p* = 0.732).

#### 3.1.2. Comparison of Vaccination Status

The vaccination status for each of the six vaccines and their combinations administered in infancy were compared between the KD and control groups in univariate analysis (Table 2). There were no significant differences between the two groups in the vaccination status of any vaccine or the combination of all vaccines, three alum-containing vaccines, and three non-alum-containing vaccines.

A conditional logistic regression analysis was performed with each vaccination and the following fixed variables: age (months), sex, breastfeeding, baby food, history of allergic disease, symptoms of infection within 2 months before hospitalization, antibiotic administration within 2 months before hospitalization, sibling(s), family history of KD, or housing style (Appendix A). We evaluated the odds ratios and their 95% CIs for the development of KD with each vaccine and found no significant association between vaccination and KD development (Table 2).

### 3.2. Case-Crossover Study

The dates of onset and vaccination were fully available for 98 of 102 cases in the KD group. Thus, these 98 cases were further subjected to the case-crossover study (Figure 1). Table 3 shows the number of vaccinated individuals, the IR_MH_, and 95% CIs for each vaccine in the case and control periods. With a case and control period of 7 days (S), we did not find evidence that vaccination was significantly more frequent in the case period. On the other hand, with a period of 28 days (L), the number of patients who received BCG vaccination was significantly lower in the case period (IR_MH_ 0.41 [0.17–0.99]). For other vaccines, no significant differences were observed between the case and control periods (Table 3). Since this analysis included multiple doses of the same vaccine, we also limited the analysis to the first dose of each vaccine and found similar results (Appendix A).

### 3.3. The Analysis of Data from the Nationwide Survey of KD in Japan

In 2012–2013, new vaccines (PCV13, Hib, and rotavirus) were added to the national vaccination schedule in Japan. Therefore, from the data of the Nationwide Survey of KD in Japan, we compared the number of patients with KD and the incidence rate of KD by age group in the first 5-year period (2007–2011) and the second 5-year period (2014–2018) (Table 4). For all ages, both the number of patients and incidence of KD were significantly higher in 2014–2018 than in 2007–2011. However, the number of KD patients younger than one year of age did not differ between the two periods. Notably, when the new vaccines were administered at 3–5 months of age, the number of cases was significantly lower in the years 2014–2018. The incidence rate at 3–5 months of age tended to be lower in 2014–2018, although it did not reach statistical significance (Table 4).

The recommended period for vaccination among infants in Japan: 13-valent pneumococcal conjugate vaccine, 2–5 months (three doses); Hepatitis B virus vaccine, 2–3 months (first and second doses) and 7–11 months (third dose); Diphtheria, pertussis, tetanus and inactivated poliovirus vaccine, 3–5 months (three doses); *Haemophilus influenzae* type b vaccine, 2–5 months (three doses); Rotavirus vaccine, 2–4.5 months (monovalent, two doses) or 2–6.5 months (pentavalent, three does); *Bacillus Calmette-Guérin* vaccine, 5–7 months (one dose).

## 4. Discussion

We have performed a case-control study considering confounding factors and a case-crossover study by analyzing the Nationwide Survey of KD. These results have shown no association between vaccination in early infancy and the development of KD.

According to an analysis of Adverse Events Following Immunization for both rotavirus vaccines based on the US Vaccine Adverse Events Reporting System (VAERS) and VigiBase, the WHO global database of Individual Case Safety Reports, the reporting odds ratio for KD was 14.61 (95% CI: 10.96–19.49) for VAERS and 5.57 (95% CI: 3.90–7.94) for VigiBase, suggesting the high association of rotavirus vaccine with KD [25]. On the other hand, a cohort study by Layton et al. [26] showed that the relative risk of developing KD in infants who received rotavirus vaccine (Rotarix^®^ or RotaTeq^®^) was 0.54 (95% CI 0.20–1.48), and there was no association between vaccination and KD development. In a meta-analysis of multiple cohort studies and RCTs, the incidence of KD was low, at 24 cases per 100,000 (95% CI: 11.98–48.26), and the relative risk of the prevalence rate was 1.55 (95% CI: 0.41–5.93), with no significant association [27].

Regarding PCV13, the relative incidence of complete KD after one dose of PCV13 was 2.59 (95% CI 1.16–5.81), which was statistically significant [28]. However, there was no significant increase in KD development after dose 2 or 3 [28]. On the other hand, the relative risk of developing KD within 28 days of vaccination with PCV13 was 2.38 (95% CI 0.92–6.38), which was not statistically significant [29]. As for the multicomponent meningococcal serogroup B (4CMenB) vaccine, the estimated incidence of KD was 72 (95% CI: 23–169) per 100,000 subject-years in the 4CMenB vaccinated group and 56 (95% CI: 1–311) in the control group, which was similar in both groups [30]. Furthermore, a systematic review of the association of various vaccinations with KD showed no evidence that vaccination increased the risk of developing KD, including the PCV13, rotavirus, and 4CMenB vaccines [31].

There are several reports that have examined the association between vaccination and the incidence of KD using self-controlled studies. Abrams et al. [32] reported that the rate of vaccination (DTaP; inactivated poliovirus; seven-valent PCV; measles, mumps and rubella, HBV; and Hib) in the 42 days before the onset of KD was significantly lower than that in the 43–84 days before the onset (control period) (rate ratio 0.79, 95% CI: 0.64–0.97). Stowe et al. [33] reported that, based on the self-controlled case-series method examining the cases in which KD developed within 28 days of vaccination, the relative incidence (RI) was 0.51 (95% CI: 0.34–0.78) for the PCV vaccine and 1.03 (95% CI: 0.54–1.98) for the meningococcal B vaccine; thus, there was no evidence that these vaccines increased the risk of KD.

Elevated serum levels of cell death-related molecules, such as S100 proteins, high mobility group box-1 protein, heat shock proteins, and oxidized phospholipids, are among the characteristics in patients with the acute phase of KD [34,35,36,37,38,39,40]. Patients with *Yersinia pseudotuberculosis* infection and severe acute respiratory syndrome coronavirus 2 (SARS-CoV-2) infection reproducibly develop principal KD symptoms fulfilling the diagnostic criteria for KD [41,42]. The induction of apoptosis and pyroptosis of endothelial cells or macrophages have been observed in patients with *Yersinia* infection [43] and coronavirus disease 2019 [44]. Pyroptosis during infection with *Yersinia* and SARS-CoV-2 releases proinflammatory cellular contents, such as damage-associated molecular patterns (DAMPs). Reactive oxygen species (ROS) immediately oxidize the membrane phospholipids of the damaged cells [45]. DAMPs, including oxidized phospholipids and low-density lipoproteins, activate the endothelial and innate immune cells to further produce proinflammatory cytokines and ROS [46]. These processes induce the activation of the NACHT, LRR, and PYD domains-containing protein 3 (NLRP3) inflammasomes, resulting in the acceleration of pyroptosis of the endothelial cells and monocytes [47]. Thus, proinflammatory cell death (pyroptosis) of endothelial and innate immune cells appears to play a critical role in developing KD vasculitis.

Vaccines include live-attenuated and non-live composite ones (Appendix A). There is no evidence of cell death associated with rotavirus vaccine (a live attenuated vaccine) or non-live Hib conjugate vaccine containing no aluminum adjuvant [48]. BCG vaccine induces the local cell death (apoptosis) of macrophages; however, BCG substrain Tokyo must have the minimum effect since it is considered the safest BCG vaccine [49,50]. With regard to non-live composite vaccines, aluminum adjuvant-containing vaccines (PCV13, DPT-IPV, and HBV vaccine) can be associated with local cell death (necrosis) at the injection site [51]. The local cell death can lead to the limited release of DAMPs (host cell components such as DNA and cell membrane). Thus, in the absence of massive proinflammatory cell death (pyroptosis) of endothelial cells and the subsequent release of excessive DAMPs, it is likely that vaccines make little contribution to the development of KD [37,52].

The present study was associated with some limitations. First, in the prospective case-control study, the numbers of patients and controls enrolled were relatively small. In addition, it was not possible to match the age and sex of the two groups perfectly. However, the timing of enrollment of the cases and the corresponding controls at each institution were almost identical. It was possible to statistically analyze the association between vaccination and KD development in the two groups by adjusting for confounding factors using multivariable analysis. Second, we conducted an epidemiological evaluation of vaccination and KD development based on questionnaires; however, we did not perform an immunological evaluation of the patients. Finally, in the case-crossover study, the long observation period was set at 8 weeks. In patients with KD at 2–3 months of age, in the control period (day -56 to -28), there were fewer opportunities to receive vaccinations than the case period. However, despite this situation, the number of vaccinations that patients with KD received in the case period did not differ from that in the control period to a statistically significant extent, which supported the present results.

## 5. Conclusions

The case-control and case-crossover studies, with the analysis of the data of the Nationwide Survey of KD, demonstrated that there was no association between vaccination in early infancy and the development of KD. Further studies of larger study populations will be useful for confirming the findings of the present study.

## Figures and Tables

**Figure 1 vaccines-09-00839-f001:**
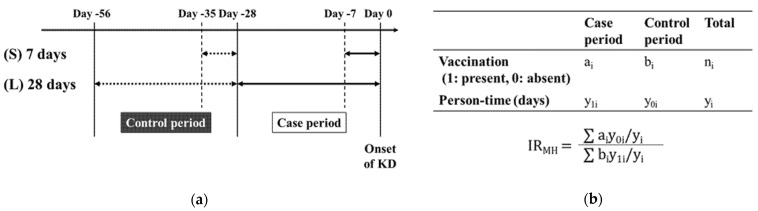
Period setting and the analysis of the case-crossover study. (**a**) Definition of the case and control periods. Case periods were defined as the 7 days (S) or 28 days (L) before the date of the onset of KD. Control periods were defined as 29 to 35 days (S) or 29 to 56 days (L) prior to the onset of KD. (**b**) A table contained the vaccination status and person-time per subject (indicating as stratum “i”) to calculate the Mantel-Haenszel incidence ratio (IR_MH_) as the pooled estimate.

**Table 1 vaccines-09-00839-t001:** Comparison of the background characteristics of the KD patients and controls.

Characteristics	KD	(*n*)	Control	(*n*)	*p*
Age (months) ^1^	6 (2–9)	(102)	5 (1–9)	(139)	0.017 ^2^
Sex (male)	64 (62.7%)	(102)	77 (55.4%)	(139)	0.29 ^3^
Breast feeding	94 (93.1%)	(101)	120 (87.6%)	(137)	0.20 ^3^
Baby food	44 (43.6%)	(101)	37 (27.0%)	(137)	0.0088 ^3^
History of allergic disease	5 (5.0%)	(100)	18 (13.0%)	(138)	0.046 ^3^
Food allergy	2 (2.0%)	(100)	9 (6.5%)	(138)	0.13 ^3^
Atopic dermatitis	3 (3.0%)	(100)	10 (7.3%)	(138)	0.25 ^3^
Asthma	0 (0%)	(100)	5 (3.6%)	(138)	0.076 ^3^
Symptoms of infection within 2 months before hospitalization	29 (29.0%)	(100)	44 (32.1%)	(137)	0.70 ^3^
Antibiotic use within 2 months before hospitalization	16 (15.7%)	(102)	21 (15.1%)	(139)	1.00 ^3^
Number of family member living together ^1^	4 (3–11)	(102)	4 (1–10)	(138)	0.28 ^2^
Sibling(s)	57 (55.9%)	(102)	84 (60.9%)	(138)	0.51 ^3^
Family history of KD	6 (6.2%)	(97)	10 (7.3%)	(137)	0.80 ^3^
Housing style (detached house)	33 (32.7%)	(101)	37 (27.6%)	(134)	0.47 ^3^

KD, Kawasaki disease. ^1^: Median (range), ^2^: Wilcoxon rank-sum test, ^3^: Fisher’s exact test.

**Table 2 vaccines-09-00839-t002:** Comparison of the vaccination status by univariate and multivariable analyses.

	KD (*n* = 102)	Control (*n* = 139)	Univariate Analysis		Multivariable Analysis	
Vaccines			Crude OR (95%CI)	*p*	Adjusted OR (95%CI)	*p*
Vaccines with alum adjuvants						
PCV13	92 (90.2%)	115 (82.7%)	1.92 (0.87–4.22)	0.10	0.98 (0.32–2.95)	0.97
HBV vaccine	89 (87.3%)	110 (79.1%)	1.80 (0.89–3.68)	0.10	0.78 (0.28–2.17)	0.64
DPT-IPV	75 (73.5%)	98 (70.5%)	1.16 (0.65–2.06)	0.60	0.39 (0.14–1.09)	0.07
Vaccines without alum adjuvants						
Hib vaccine	93 (91.2%)	115 (82.7%)	2.15 (0.95–4.86)	0.06	1.19 (0.37–3.85)	0.77
Rotavirus vaccine	64 (62.7%)	74 (53.2%)	1.47 (0.88–2.49)	0.14	1.32 (0.62–2.77)	0.47
BCG vaccine	48 (47.1%)	48 (34.5%)	1.68 (0.99–2.84)	0.05	0.58 (0.17–1.96)	0.38

The univariate analysis was performed using univariate logistic regression analysis. The multivariable analysis was performed using conditional logistic regression analysis. KD: Kawasaki disease, OR: Odds ratio, 95%CI: 95% confidence interval, PCV13: 13-valent pneumococcal conjugate vaccine, HBV: Hepatitis B virus, DPT-IPV: Diphtheria, pertussis, tetanus and inactivated poliovirus vaccine, Hib: *Haemophilus influenzae* type b, BCG: *Bacillus Calmette-Guérin*.

**Table 3 vaccines-09-00839-t003:** Case-crossover study on vaccination in patients with KD.

Vaccines	Number of Subjects Who Received Vaccine in the Case Period	Number of Subjects Who Received Vaccine in the Control Period	IR_MH_	(95%CI)
(S) 7 days				
PCV13	9	9	1.00	(0.40–2.52)
HBV vaccine	7	6	1.17	(0.39–3.47)
DPT-IPV	7	5	1.40	(0.44–4.41)
Hib vaccine	9	8	1.13	(0.43–2.92)
Rotavirus vaccine	4	2	2.00	(0.37–10.92)
BCG vaccine	2	3	0.67	(0.11–3.99)
(L) 28 days				
PCV13	29	33	0.88	(0.53–1.45)
HBV vaccine	24	27	0.89	(0.51–1.54)
DPT-IPV	20	30	0.67	(0.37–1.17)
Hib vaccine	29	33	0.88	(0.53–1.45)
Rotavirus vaccine	15	23	0.65	(0.34–1.25)
BCG vaccine	7	17	0.41	(0.17–0.99)

IR_MH_: Mantel-Haenszel incidence ratio, 95%CI: 95% confidence interval, PCV13: 13-valent pneumococcal conjugate vaccine, HBV: Hepatitis B virus, DPT-IPV: Diphtheria, pertussis, tetanus and inactivated poliovirus vaccine, Hib: *Haemophilus influenzae* type b, BCG: *Bacillus Calmette-Guérin.*

**Table 4 vaccines-09-00839-t004:** The number of KD patients and incidence of KD by age group for the 5-year periods before and after 2012–2013 according to the Nationwide Survey of KD.

	2007–2011	2014–2018	*p*
Number of patients			
Total of all ages	11,970 (775)	16,020 (375)	<0.0001
<1 year old	3073 (180)	3161 (111)	0.38
0–2 months	235 (9)	240 (9)	0.68
3–5 months	839 (52)	702 (13)	0.0004
6–8 months	997 (66)	1013 (57)	0.70
9–11 months	1001 (86)	1206 (62)	0.0025
Incidence rate ^1^			
Total of all ages	224.6 (16.0)	324.0 (21.6)	<0.0001
0–2 months	87.8 (7.0)	99.9 (9.1)	0.046
3–5 months	313.4 (21.9)	291.4 (10.8)	0.078
6–8 months	372.0 (24.6)	420.6 (28.6)	0.021
9–11 months	373.8 (32.2)	500.6 (20.7)	< 0.0001

Mean (standard deviation). ^1^: Based on the population data in Vital Statistics of Japan per 100,000 populations per year.

## Data Availability

The data published in this study are available upon request to the corresponding author. This data has not been made public to protect privacy.

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
