# Peer review of "Kawasaki Disease and Vaccination: Prospective Case-Control and Case-Crossover Studies among Infants in Japan"

_vaccines, 2021, doi:10.3390/vaccines9080839_

Round 1

Reviewer 1 Report

This study deals with a very interesting and very topical topic both as regards the ability of vaccines to trigger KD, and as regards future vaccinations after IVIG administration.

I also believe that the evaluation of possible confounding elements and the simultaneous evaluation of cases and controls was important  too.

Author Response

We appreciate the reviewer 1’s comments.

Reviewer 2 Report

The paper is very interesting; however, the following concerns should be addressed.

The rationale for the study is unclear as the introduction is a bit confusing, comprising several pieces of apparently unlinked information. A more integrated appraisal of the relevant literature would be appropriate to provide the context for the study.

The following reports should be mentioned:

PMID: 34071896 

PMID: 32403217 

PMID: 32750108 

PMID: 32638151 

PMID: 31428922

PMID: 31200965

PMID: 27212174 

Author Response

We appreciate the reviewer’s comments. We have revised the introduction as follows to make the rationale clearer, and added reference 9 (page 2, lines 50-56).

“Although numerous microbes have been proposed as etiologic agents of KD, few have been consistently associated with the illness [4]. Similarly, there have been reports of patients developing KD after receiving rotavirus vaccine, pneumococcal conjugate vaccine (PCV), hepatitis B virus (HBV) vaccine, diphtheria, pertussis, tetanus and inactivated poliovirus vaccine (DPT-IPV), Haemophilus influenzae type b (Hib) vaccine and Bacillus Calmette-Guérin (BCG) vaccine [8]. Among them, the rotavirus vaccine RotaTeq® was associated with a higher risk of KD in pre-marketing phase 3 clinical trials, and KD was then listed as one of the serious side effects of RotaTeq® in 2007 [9].”

Among the references suggested by the reviewer 2, PMID: 32750108 is cited as reference 44 in lines 308 and 482-485, and PMID: 31428922 is cited as reference 46 in lines 313 and 488-490.

  1. Evans, P.C.; Rainger, G.E.; Mason, J.C.; Guzik, T.J.; Osto, E.; Stamataki, Z.; Neil, D.; Hoefer, I.E.; Fragiadaki, M.; Waltenberger, J.; et al. Endothelial dysfunction in COVID-19: a position paper of the ESC Working Group for Atherosclerosis and Vascular Biology, and the ESC Council of Basic Cardiovascular Science. Cardiovascular research 2020, 116, 2177-2184, doi:10.1093/cvr/cvaa230.
  2. He, Y.E.; Qiu, H.X.; Wu, R.Z.; Rong, X.; Xu, H.T.; Xiang, R.L.; Chu, M.P. Oxidised Low-Density Lipoprotein and Its Receptor-Mediated Endothelial Dysfunction Are Associated with Coronary Artery Lesions in Kawasaki Disease. J Cardiovasc Transl Res 2020, 13, 204-214, doi:10.1007/s12265-019-09908-y.

The following four references were not related to the contents of our manuscript. We do not think they will improve our manuscript.

PMID: 34071896; Chang LS, Yu HR, Chu CL, Chen KD, Huang YH, Guo MM, Weng KP, Kuo HC. Long-Term Hypermethylation of FcγR2B in Leukocytes of Patients with Kawasaki Disease. J Clin Med. 2021 May 27;10(11):2347. doi: 10.3390/jcm10112347.

PMID: 32403217; Sardu C, Gambardella J, Morelli MB, Wang X, Marfella R, Santulli G. Hypertension, Thrombosis, Kidney Failure, and Diabetes: Is COVID-19 an Endothelial Disease? A Comprehensive Evaluation of Clinical and Basic Evidence. J Clin Med. 2020 May 11;9(5):1417. doi: 10.3390/jcm9051417.

PMID: 31200965. Ohnishi Y, Yasudo H, Suzuki Y, Furuta T, Matsuguma C, Azuma Y, Miyake A, Okada S, Ichihara K, Ohga S, Hasegawa S. Circulating endothelial glycocalyx components as a predictive marker of coronary artery lesions in Kawasaki disease. Int J Cardiol. 2019 Oct 1;292:236-240. doi: 10.1016/j.ijcard.2019.05.045. Epub 2019 May 22.

PMID: 27212174. Tian J, Lv HT, An XJ, Ling N, Xu F. Endothelial microparticles induce vascular endothelial cell injury in children with Kawasaki disease. Eur Rev Med Pharmacol Sci. 2016 May;20(9):1814-8.

We did not cite PMID: 32638151, because we do not agree with the description that antigen–antibody immune complexes can start inflammatory type III hypersensitivity symptoms, including protease releases that can disrupt epithelium, mesothelium, and endothelium basement membranes, and induce pervasive inflammation in KD. Extensive efforts have been made in search for the deposit of immune complex at vascular lesions of KD in Japan and other countries. However, no immune complex depositions have been detected in KD vasculitis lesions by Japanese and American pathologists (1-3).

References

  1. Menikou S, Langford PR, Levin M. Kawasaki Disease: The Role of Immune Complexes Revisited. Front Immunol 2019; 10: 1156.
  2. Rowley AH, Eckerley CA, Jäck HM, Shulman ST, Baker SC. IgA plasma cells in vascular tissue of patients with Kawasaki syndrome. J Immunol 1997;159:5946-55.
  3. Takahashi K, Oharaseki T, Yokouchi Y, Hiruta N, Naoe S. Kawasaki disease as a systemic vasculitis in childhood. Ann Vasc Dis 2010;3:173-81.

PMID: 32638151; Roe K. A viral infection explanation for Kawasaki disease in general and for COVID-19 virus-related Kawasaki disease symptoms. Inflammopharmacology. 2020 Oct;28(5):1219-1222. doi: 10.1007/s10787-020-00739-x. Epub 2020 Jul 7.

Reviewer 3 Report

I have read the article by Murata et al. with great interest. The authors evaluated the potentials relationship between vaccinations in infancy and the development of Kawasaki disease. I think this is a well written paper with a concordant conclusion obtained from 3 sub-studies.

I have only one major comments:

  • 2.1.3. Please provide power calculations as it is critical to understand if the study has reached the minimal sample size for calculations. Should this show that the sample size was a limitation then please discuss it.

Author Response

We appreciate the reviewer’s comment on this point.

As described in the background, this study initially began with a review of rotavirus vaccines, which have been widely reported to be associated with Kawasaki disease. During the study period, rotavirus vaccine was a voluntary vaccination in Japan. In a retrospective study conducted at our institution, the rotavirus vaccine coverage was 39% in the KD group and 21% in the control group. Therefore, when we calculated the sample size based on those proportions with a case-control ratio 1:1, 80% power and a two-tailed alpha of 0.05, the minimum sample size was 100 cases in each group.

Reviewer 4 Report

Major:

1.Most research Description that Kawasaki disease (KD) occurred mostly in infants and children younger than 5 years of age and the age-specific incidence rate according to sex was highest in children between 9 and 11 months of age like [PMID: 30786118], However, this research he case-control study included infants of 6 weeks to 9 months of age. Please explain the reason for the choice

2.According to the study design fromline 97-98, the collection of data in the case-control study were from questionnaires based on retrospective It should be proved further that the study is prospective.   

3.Considered the risk of weight, it is suggested that the patients can be classified into different groups by body weight to compare the background characteristics of the KD patients and controls. [PMID: 33812731]

4.The method of the statistical analyses mentioned in the line 128-133 should be more detailed such as describing the Wilcoxon rank-sum test instead of ANOVA used for continuous variables.

5.In the method and result, the questionnaire returns including valid questionnaires and missing data should be described.

6.In the multicenter study, it is confused that the central tendency of age was described by the median in the condition of KD group, n=102, and control group, n=139.

Minor:

1.Figure 1a, the abscissa is not displayed completely ,it should be shown completed.

2.The references should be cited for the sentences in lines 52-53.And,Some articles demonstrate the associations between Kawasaki disease and SARS-Cov-2 [PMID: 33732254], Authors need to cite the latest references instead of references 40-41.

3.The grammar is modified to “admit to participating” in line 78.

4.In table 1, the meaning of the sign † was missing.

Author Response

We appreciate the reviewer's valuable comments.

Major:

1. Most research Description that Kawasaki disease (KD) occurred mostly in infants and children younger than 5 years of age and the age-specific incidence rate according to sex was highest in children between 9 and 11 months of age like [PMID: 30786118], However, this research he case-control study included infants of 6 weeks to 9 months of age. Please explain the reason for the choice

In Japan, vaccines are most frequently administered in pre- and mid-infancy. The recommended time periods for vaccination are 2-4 months for monovalent rotavirus vaccines, 2-6.5 months for pentavalent rotavirus vaccines, 2-4 months for the first and second doses of HBV vaccine, 2-6 months for Hib and PCV vaccines, 3-6 months for DPT-IPV vaccine, and 5-8 months for BCG. Since new vaccines (rotavirus, PCV13 and Hib) were added to the national vaccination schedule in 2012-2013 in Japan, and the incidence of KD continued to increase until 2018, the study population was set from 6 weeks to 9 months of age.

Lines 75-80 were modified as follows:

“This multicenter prospective case-control study included infants of 6 weeks to 9 months of age who were admitted to the seven institutions with KD (case) or without KD (control) between January 1, 2017 and June 30, 2019. Since new vaccines (rotavirus, PCV13 and Hib) were added to the national vaccination schedule in 2012-2013 in Japan, and the incidence of KD continued to increase until 2018, the study population was set from 6 weeks to 9 months of age.”

2. According to the study design from line 97-98, the collection of data in the case-control study were from questionnaires based on retrospective It should be proved further that the study is prospective.  

As Reviewer 4 pointed out, the term “nested case-control study” may be more widely accepted for this study design than “prospective case-control study.” However, as shown in below (4-5), there are many papers that used the term “prospective case-control study” for the similar design as ours. We used the term “prospective” in our manuscript because most of the previous studies recruit patients retrospectively and we believe that the prospective enrollment of the cases and controls is one of the strong points of the present study.

References

  1. Bénet T, et al. Microorganisms Associated With Pneumonia in Children <5 Years of Age in Developing and Emerging Countries: The GABRIEL Pneumonia Multicenter, Prospective, Case-Control Study. Clin Infect Dis 2017;65(4):604-612.
  2. Maritsi DN, et al. Antibody status against measles in previously vaccinated childhood systemic lupus erythematosus patients: a prospective case-control study. Rheumatology (Oxford) 2018;57(8):1491-1493.

3. Considered the risk of weight, it is suggested that the patients can be classified into different groups by body weight to compare the background characteristics of the KD patients and controls. [PMID: 33812731]

As pointed out by the reviewer, weight may be a relevant factor, but it was not included in the questionnaire and is difficult to evaluate.

4. The method of the statistical analyses mentioned in the line 128-133 should be more detailed such as describing the Wilcoxon rank-sum test instead of ANOVA used for continuous variables.

In order to describe more detailed methods of statistical analysis, lines 129-135 were modified as follows:

“Continuous variables were tested for normality using the Shapiro-Wilk test. Comparison between the two groups was performed using the Wilcoxon rank sum test because some variables were not normally distributed. The differences in categorical variables were compared between the two groups using Fisher’s exact test. The classifications of the parents’ occupations were compared between the two groups using the Pearson’s chi-squared test. A univariate logistic regression analysis was used to examine the association between each vaccine and the development of KD and to calculate crude odds ratios.”

5. In the method and result, the questionnaire returns including valid questionnaires and missing data should be described.

We were able to collect 100% of the questionnaires from the subject's physicians and parents who agreed to participate in the study. Eighteen patients (KD group, n=8; control group, n=10) had some missing data in the questionnaire items, but vaccination history was collected without missing from all 241 patients.

Lines 180-184 were modified as follows:

“The subjects included 241 patients (KD group, n=102; control group, n=129) from seven participating institutions who met the enrollment criteria, and questionnaires could be collected from the subject's physician and parents in all cases. Eighteen patients (KD group, n=8; control group, n=10) had some missing data in the questionnaire items (Table 1), but vaccination history was collected without missing from all 241 patients.”

6. In the multicenter study, it is confused that the central tendency of age was described by the median in the condition of KD group, n=102, and control group, n=139.

The median age in months for both groups at each institution is shown in the table below. The median age was equal or higher in the KD group at all institutions, and there was no obvious difference between the institutions.

Institutions

KD

[N]

Control

[N]

P

A

6 (2-9)

42

5 (1-9)

78

0.19

B

6 (3-9)

24

4 (2-9)

24

0.28

C

5 (2-8)

18

5 (2-8)

18

0.91

D

7 (3-9)

9

4 (2-9)

10

0.10

E

6 (4-9)

5

5 (2-8)

5

0.46

F

8 (7-9)

2

5 (1-8)

2

0.44

G

9 (9-9)

2

9 (9-9)

2

1.00

Total

6 (2-9)

102

5 (1-9)

139

0.017

Minor:

1. Figure 1a, the abscissa is not displayed completely, it should be shown completed.

As pointed out by the reviewer, the misplaced part of Figure 1a has been corrected.

2.The references should be cited for the sentences in lines 52-53.And,Some articles demonstrate the associations between Kawasaki disease and SARS-Cov-2 [PMID: 33732254], Authors need to cite the latest references instead of references 40-41.

The following reference #9 has been added to lines 56 and 397-398:

  1. U.S. Food and Drug Administration. Package Insert - RotaTeq. Available online: https://www.fda.gov/vaccines-blood-biologics/vaccines/rotateq (accessed on October 13, 2020).

Next, we added the new reference #41 [PMID: 26561332] and the recommended reference #42 [PMID: 33732254] that the reviewer pointed out regarding KD and SARS-CoV-2 to lines 315 and 475-479.

  1. Horinouchi, T.; Nozu, K.; Hamahira, K.; Inaguma, Y.; Abe, J.; Nakajima, H.; Kugo, M.; Iijima, K. Yersinia pseudotuberculosis infection in Kawasaki disease and its clinical characteristics. BMC pediatrics 2015, 15, 177, doi:10.1186/s12887-015-0497-2.
  2. Chen, M.R.; Kuo, H.C.; Lee, Y.J.; Chi, H.; Li, S.C.; Lee, H.C.; Yang, K.D. Phenotype, Susceptibility, Autoimmunity, and Immunotherapy Between Kawasaki Disease and Coronavirus Disease-19 Associated Multisystem Inflammatory Syndrome in Children. Front Immunol 2021, 12, 632890, doi:10.3389/fimmu.2021.632890.

In addition, we cited two new references #43 [PMID: 25879289] and #44 [PMID: 32750108] instead of the original 40-41 references to lines 307-308 and 480-485.

  1. Jorgensen, I.; Miao, E.A. Pyroptotic cell death defends against intracellular pathogens. Immunol Rev 2015, 265, 130-142, doi:10.1111/imr.12287.
  2. Evans, P.C.; Rainger, G.E.; Mason, J.C.; Guzik, T.J.; Osto, E.; Stamataki, Z.; Neil, D.; Hoefer, I.E.; Fragiadaki, M.; Waltenberger, J.; et al. Endothelial dysfunction in COVID-19: a position paper of the ESC Working Group for Atherosclerosis and Vascular Biology, and the ESC Council of Basic Cardiovascular Science. Cardiovascular research 2020, 116, 2177-2184, doi:10.1093/cvr/cvaa230.

3. The grammar is modified to “admit to participating” in line 78.

Reviewer 4 may have misinterpreted that “patients admitted to participating”, but this sentence means that “patients were admitted to the seven participating hospitals”. As it might be confusing, Line 76 has been modified as follows:

“This multicenter prospective case-control study included infants of 6 weeks to 9 months of age who were admitted to the seven institutions with KD (case) or without KD (control) between January 1, 2017 and June 30, 2019.”

4. In table 1, the meaning of the sign † was missing.

We appreciate you pointing this out. The † has been corrected to footnote “2”.

Round 2

Reviewer 4 Report

This study was designed to investigae the association between vaccines administered during infancy and the development of KD in Japan. However, there are still some minor problems in this article. 

1. The authors aimed to examine the association between vaccines administered during infancy and the development of KD in Japan by a multicenter prospective case-control study and a case-control study. However, some published studies are on the same topic (PMID:31265459, 32736940, 27651105), so the innovation of this study is limited.

2. In the summary and introduction part of this study, the authors could add the contents related to the hazards of KD to highlight the clinical value of the purpose of this study.

3. Due to the lack of vaccine-related data, a statistical analysis of the data from the Nationwide Survey of KD in Japan cannot reveal an association between vaccination and the development of KD.

4. This article is lacking of inclusion criteria.

5. In lines 248 to 249 of the article, the text description is inconsistent with Table 4. Please check carefully.

6. It is better to make “ P ” value be bold if is statistical significance in all tables.

Author Response

We again appreciate the reviewer's insightful comments.

1. The authors aimed to examine the association between vaccines administered during infancy and the development of KD in Japan by a multicenter prospective case-control study and a case-control study. However, some published studies are on the same topic (PMID:31265459, 32736940, 27651105), so the innovation of this study is limited.

As the reviewer commented, there are several reports that examine the relationship between Kawasaki disease and vaccination. However, the present study is a prospective case-control study matched for region and time period, taking into account the effects of confounding factors, and this approach to analysis is unique to previous reports. In addition, the multifaceted evaluation using multiple analysis methods is the strength of this study, and we believe that the study is sufficiently worthy of reporting.

2. In the summary and introduction part of this study, the authors could add the contents related to the hazards of KD to highlight the clinical value of the purpose of this study.

As the Reviewer pointed out, the sequelae of Kawasaki disease include giant coronary aneurysms and coronary artery stenosis, leading to myocardial infarction and death. Lines 47-48 in Introduction were modified as follows:

“Coronary artery abnormalities develop in 15–25% of untreated children with KD, and the sequelae include giant coronary aneurysms and coronary artery stenosis, leading to myocardial infarction and death [2].”

3. Due to the lack of vaccine-related data, a statistical analysis of the data from the Nationwide Survey of KD in Japan cannot reveal an association between vaccination and the development of KD.

As this reviewer points out, the nationwide survey of KD in Japan does not include any data on vaccination, thus it is impossible to assess a direct association between vaccination and the development of KD. However, we consider that it is possible to evaluate the indirect association between vaccination and KD by comparing two periods in which vaccination rates are clearly different.

4. This article is lacking of inclusion criteria.

We described the inclusion criteria in 2 separate parts. So, we combined them into one. As described in lines 77-86, this study included all infants of 6 weeks to 9 months of age who were admitted to the seven institutions with KD (case) or without KD (control) between January 1, 2017 and June 30, 2019, excluding those who met the exclusion criteria.

“2.1.1. Study design and subjects

This multicenter prospective case-control study included infants of 6 weeks to 9 months of age who were admitted to the seven institutions with KD (case) or without KD (control) between January 1, 2017 and June 30, 2019. Since new vaccines (rotavirus, PCV13 and Hib) were added to the national vaccination schedule in 2012-2013 in Japan, and the incidence of KD continued to increase until 2018, the study population was set from 6 weeks to 9 months of age. KD was diagnosed according to the diagnostic guidelines for Kawasaki disease (5th revised edition) of Japan [11]. Cases included both complete KD (five or six of six major symptoms) and incomplete KD (four of six major symptoms with coronary artery aneurysm). Controls were selected from patients who were admitted to the same facility at the same time as the KD patients.”

5. In lines 248 to 249 of the article, the text description is inconsistent with Table 4. Please check carefully.

The "Total" in Table 4 indicates the total of all ages, not the sum of items below it. To make it obvious, we changed "Total" in Table 4 to "Total of all ages".

6. It is better to make “ P ” value be bold if is statistical significance in all tables.

As the Reviewer pointed out, we modified the P-values in Table to bold.

7. English language and style are fine/minor spell check required

Dr. Brian Quinn (Japan Medical Communication, Fukuoka, Japan) checked English language and style of the manuscript.
